

# Aerosol size distributions during the Atmospheric Tomography (ATom) mission: methods, uncertainties, and data products

Charles A. Brock[1], Christina Williamson[1,2], Agnieszka Kupc[1,2], Karl Froyd[1,2], Frank Erdesz[1,2], Nicholas Wagner[1,2], Matthews Richardson[1,2], Joshua P. Schwarz[1], Ru-Shan Gao[1], Joseph M. Katich[1,2], Pedro Campuzano-Jost[2,3], Benjamin A. Nault[2,3], Jason C. Schroder[2,3], Jose L. Jimenez[2,3], Bernadett Weinzierl[4], Maximillian Dollner[4], ThaoPaul Bui[5] and Daniel M. Murphy[1]

[1]NOAA Earth System Research Laboratory, Boulder, 80305, U.S.A.
[2]Cooperative Institute for Research in Environmental Sciences, University of Colorado, Boulder, 80309, U.S.A.
[3]Department of Chemistry, University of Colorado, Boulder, 80309, U.S.A.
[4]Faculty of Physics, Aerosol Physics and Environmental Physics, University of Vienna, 1090 Wien, Austria
[5]Atmospheric Science Branch, NASA Ames Research Center, Moffett Field, 94035, U.S.A.

*Correspondence to*: Charles A. Brock (charles.a.brock@noaa.gov)

**Abstract.** From 2016-2018 a DC-8 aircraft operated by the U.S. National Aeronautics and Space Administration (NASA) made four series of flights, profiling the atmosphere from 150 m to ~12 km above sea level from the Arctic to the Antarctic over both the Pacific and Atlantic Oceans. This program, the Atmospheric Tomography (ATom) mission, sought to sample the troposphere in a representative manner, making measurements of atmospheric composition in each season. This paper describes the aerosol microphysical measurements and derived quantities obtained during this mission. Dry size distributions from 2.7 nm to 4.8 µm in diameter were measured in-situ at 1 Hz using a battery of instruments: 10 condensation particle counters with different nucleation diameters, two ultra-high sensitivity aerosol size spectrometers (UHSAS), one of which measured particles surviving heating to 300° C, and a laser aerosol spectrometer (LAS). The dry aerosol measurements were complemented by size distribution measurements from 0.5-930 µm diameter at near-ambient conditions using a cloud, aerosol, and precipitation spectrometer (CAPS) mounted under the wing of the DC-8. Dry aerosol number, surface area, and volume, and optical scattering and asymmetry parameter at several wavelengths from the near-UV to the near-IR were calculated from the measured dry size distributions (2.7 nm to 4.8 µm). Dry aerosol mass was estimated by combining the size distribution data with particle density estimated from independent measurements of aerosol composition with a high-resolution aerosol mass spectrometer and a single particle soot photometer. This paper briefly describes the instrumentation and fully documents the aircraft inlet and flow distribution system, the derivation of uncertainties, and the calculation of data products from combined size distributions. Comparisons between the instruments and direct measurements of some aerosol properties confirm that in-flight performance was consistent with calibrations and within stated uncertainties for the two deployments analyzed. The unique ATom dataset contains accurate, precise, high-resolution in-situ measurements of dry aerosol size distributions, and integral parameters, and estimates and measurements of optical properties, for particles <4.8 µm in diameter that can be used to evaluate aerosol abundance and processes in global models.

## 1 Introduction

The Atmospheric Tomography Mission (ATom) was designed to improve understanding of chemistry and climate processes in the remote atmosphere over the oceans (https://espo.nasa.gov/home/atom). Using the long-range NASA DC-8 research aircraft, ATom consisted of four series of flights over the middle of the Pacific and Atlantic Oceans, spanning from ~82° N to ~86° S latitude (Fig. 1). During these flights, the DC-8, repeatedly ascended and descended between ~0.15 and ~12 km in altitude. The four flight series were completed in August-September 2016 (ATom-1), January-February 2017 (ATom-2), September-October 2017 (ATom-3), and April-May 2018 (ATom-4). ATom's stated scientific goals included determining the



abundance and distribution of aerosol components in the remote troposphere, identifying the sources of these particles, and evaluating mechanisms of new particle formation and growth of newly formed particles to cloud condensation nuclei (CCN) sizes. The datasets produced by the ATom project are expected to provide extensive and unique constraints on global models that simulate atmospheric chemistry and climate; thus, it is important to explicitly document the measurement methodology,

the resulting data products, and their limitations and uncertainties.

This manuscript briefly describes the instruments used to measure size distributions on the NASA DC-8 aircraft during the ATom flights while thoroughly discussing the sampling system, the data processing and uncertainty calculations, and the available data products. The focus is on dry measurements of the nucleation and accumulation modes, those particles with diameter ($D_p$) from 3-500 nm which are most relevant to new particle formation and aerosol-cloud interactions, and of the

portion of the coarse mode <4.8 μm in diameter that is efficiently sampled through the aircraft inlet. The portion of the coarse mode between 4.8 and 50 μm, which can dominate total particle mass in the marine boundary layer (MBL) and in strong dust plumes, was measured using underwing probes at ambient relative humidity conditions and is not described here. Detailed descriptions of individual instruments and their calibrations are not included since they have already been published. .

## 2 Instrumentation

Several instruments–a nucleation-mode aerosol size spectrometer (NMASS), an ultra-high sensitivity aerosol size spectrometer (UHSAS), a laser aerosol spectrometer (LAS), a printed optical particle sizer (POPS) and a second-generation cloud, aerosol and precipitation spectrometer (CAPS)–were used to measure the size distribution from 3 nm to 930 μm. All instruments, except the CAPS, sampled continuously from an inlet developed by the University of Hawaii (UH) and now operated by the NASA Langley Aerosol Research Group (LARGE). The inlet and sampling system are described in Sect. 3.

### 2.1 Nucleation-mode aerosol size spectrometer (NMASS)

The NMASS is a custom-built battery of five continuous-laminar-flow condensation particle counters (CPCs) operating in parallel within a single instrument (Brock et al. 2000; Williamson et al. 2018). Each CPC module, or channel, is operated at a different condenser temperature and thus nucleates and grows a different size class of particles to droplets, which are optically counted. During ATom-1, a single NMASS was operated at fixed size cuts of 3.2, 8.3, 14, 27, and 59 nm, representing the $D_p$

at which each channel detects particles with >50 % efficiency. During ATom-2-4, a second NMASS instrument was operated in parallel with the first, providing additional size cuts at 5.2, 6.9, 11, 20, and 38 nm, resulting in a total of 10 size classifications with 1-s time resolution. Each NMASS instrument operates at a fixed internal pressure of 120 hPa behind one of two automatically switched orifices of 500 or 750 μm diameter. The smaller orifice is used at upstream pressures >425 hPa to reduce pumping requirements, and the larger one at lower pressures to allow sufficient volumetric flow in the instrument. The

working fluid for the NMASS is perfluorotributylamine (Fluorinert FC-43, 3M, St. Paul, Minnesota, USA), which offers thermodynamic advantages at 120 hPa compared to the more-commonly-used water or butanol working fluids. The NMASS instruments were calibrated as described in detail in Williamson et al. (2018). The sizing performance of the 10 CPC channels is shown in Fig. 2.

### 2.2 Ultra-high sensitivity aerosol size spectrometer (UHSAS)

A UHSAS instrument (laboratory model, Droplet Measurement Technologies, Longmont, Colorado, U.S.A.) was operated during ATom to measure with >90 % counting efficiency the concentration of particles from 63-1000 nm as a function of diameter. During ATom-1, a second UHSAS was operated in parallel with the primary instrument and proved highly valuable in assessing the instrumental uncertainties (Kupc et al., 2018). During ATom-2 and -3, a 300° C thermodenuder was installed upstream of the detector of the second UHSAS to volatilize refractory components. The thermodenuded UHSAS was not



operated during ATom-4; it was replaced by an aerosol optical extinction and absorption instrument (Sect. 2.4). Details of flow modifications to the UHSASes, calibrations of the sizing and counting efficiency, and thermodenuder performance are detailed in Kupc et al. (2018). Briefly, the UHSAS instruments were calibrated using polystyrene latex (PSL) microspheres and by using a differential mobility analyzer to size-select atomized ammonium sulfate, di-2ethylhexyl sebacate, and the

condensation products of limonene ozonolysis. Sizing uncertainties caused by the range of refractive index found in typical non-absorbing atmospheric particles are likely bounded by the range of real refractive indices in these laboratory compounds, from 1.44 to 1.58 at the UHSAS laser wavelength of 1053 nm, comparable to the refractive indices of dilute sulfuric acid and sodium chloride, respectively. Particle size distributions are reported using the ammonium sulfate calibrations, which are near the middle of this refractive index range (real refractive index 1.53; Hand and Kreidenweis, 2002).

During the ATom flights, four PSL sizes–81, 125, 240, and 400 nm–were used to make systematic, repeated calibration checks of the UHSAS, LAS (Sect. 2.3) and POPS (Sect. 2.4) instruments. During test flights, prior to deployment, these PSL particles were atomized with a portable air compressor and nebulizer and aspirated into the inlet plenum while in flight. The sizing performance of the UHSAS instruments did not change as a function of altitude during these in-flight tests (Kupc et al., 2018). Subsequently, PSL calibration checks were made on the ground before and after each flight to avoid interrupting in-flight

measurements. Additional complete calibrations using ammonium sulfate were performed at the beginning and ending of each ATom deployment, using the instrument package and plumbing as configured for flight in the aircraft. Checks of instrument zeros, using a high-efficiency particulate arrestance (HEPA) filter on the inlet plenum, were made at altitudes >8 km for ~5 minutes at least once during each flight. The sizing performance of the primary UHSAS instrument was remarkably stable, with no trend and a standard deviation of reported size for the PSL standards of <1.2 % throughout the ATom-1 and -2

deployments (Kupc et al., 2018).

The thermodenuder on the second UHSAS inlet was tested over a range of particle compositions and sizes (Kupc et al., 2018). The thermodenuder effectively evaporated ammonium sulfate particles up to $D_p$=0.5 µm at temperatures <230 ºC, while sodium chloride particles did not volatilize detectably at temperatures up to 310ºC. The thermodenuder was operated at 300ºC in flight. The thermodenuder employs an activated carbon liner to prevent recondensation of particles following evaporation,

and no formation of particles from recondensation was evident during calibrations.

### 2.3 Laser aerosol spectrometer (LAS)

A LAS (Model 3340, TSI Inc, St. Paul, Minnesota, U.S.A.) measured with high counting efficiency the size distribution from 120 nm to 10 µm diameter. The largest detectable size of the instrument was effectively limited to <4.8 µm by the inlet passing efficiency (Sect. 3). The optical geometry of the LAS is very similar to the UHSAS (they were originally developed by the

same manufacturer). The flow system of the LAS was modified to be similar to those of the UHSAS instruments (Kupc et al., 2018) so that the sample volumetric flow rate could be directly measured. The LAS was calibrated with ammonium sulfate, DES, and PSL particles. As with the UHSAS instruments, particle concentrations are reported as a function of optically equivalent diameter based on the refractive index of ammonium sulfate. These reported diameters were calculated by simulating the theoretical optical response of the instrument with Mie theory and using the calibrations with PSL particles to

scale detector voltage levels to scattering intensity.

The LAS performed well during ATom-1. However, during the second ATom deployment, an optical window was cracked and the instrument was disassembled for repairs shortly before the deployment. Upon re-assembly and testing, the LAS was found to leak into the sheath flow that surrounds the inlet flow into the detector. Although cabin particles did not contaminate the measurements, the increased sheath flow led to lower detection efficiency. Altitude-dependent corrections to the LAS were

developed based on comparison to other instruments. For this deployment only, the UHSAS was used for particles with $D_p$ <0.97 µm, and corrected LAS data over the size range 0.97 µm $\leq D_p$ < 4.8 µm. The LAS was repaired and extensively tested prior to ATom-3.





### 2.4 Other aerosol instruments

Other aerosol instruments were operated on the DC-8 during the ATom flights. A printed optical particle sizer (POPS) instrument (Gao et al., 2016) was operated as a back-up coarse mode particle counter. This simple, lightweight instrument, designed for operation on balloon and unpiloted airborne vehicles, uses a light-emitting-diode source to optically count particle

concentrations as a function of diameter from 0.14-5 µm. The instrument operated beginning with the second ATom deployment. Data from the POPS, which has lower size resolution than the LAS, were used to develop corrections for the LAS size distributions during this deployment.

The particle analysis by laser mass spectrometry (PALMS) instrument is a single-particle laser-ablation/ionization aerosol mass spectrometer (Thomson et al., 2000; Murphy et al., 2006). This instrument measures single particle composition from

~0.18-5 µm, limited at large diameters by inlet transmission efficiency. Particle composition statistics, rather than mass concentrations, are reported by the PALMS as a function of particle size for most particle compositions. These size-dependent compositional statistics are then mapped to the size distribution reported by the UHSAS and LAS, providing an estimate of the mass contribution of each particle type as a function of diameter (Froyd et al., manuscript in preparation, 2019). The mass fractions of sulfate and organic compounds in individual particles can be quantitatively determined (Murphy et al., 2006), as

can the degree of sulfate neutralization in dry particles (Froyd et al., 2009).

A single-particle soot photometer (SP2, Droplet Measurement Technologies, Longmont, Colorado, U.S.A.; Schwarz et al., 2010a) was also operated during ATom. The SP2 instrument provides a quantitative measurement of the refractory black carbon (rBC) mass of individual particles in the accumulation mode, as well as information about the mixing state of rBC-containing particles. This is the same SP2 detection system that was operated on the HIAPER Pole-to-Pole Observations

(HIPPO) missions (Schwarz et al., 2010b), with a detection range for rBC mass in the range 90 - 550 nm volume-equivalent diameter assuming 1.8 kg m$^{-3}$ void-free density for rBC .

A high-resolution time-of-flight aerosol mass spectrometer (HR-ToF-AMS; DeCarlo et al., 2006; Canagaratna et al., 2007; Schroder et al., 2018) was operated by the University of Colorado, Boulder, on the ATom flights. The HR-ToF-AMS measured the composition of particles volatile at ~600 ºC with 1 s time resolution over a diameter range from ~0.02 to ~0.7 µm geometric

diameter, with detection efficiencies >50 % between ~0.04 and ~0.5 µm. The HR-ToF-AMS used an inlet (HIMIL, Stith et al., 2009) located aft of the UH/LARGE inlet. Inlet height was raised by 10 cm to place the inlet tip clearly outside the boundary layer of the plane (Vay et al, 2003). Once each flight in ATom-1, and less frequently during ATom-2 and subsequent deployments, the sample line for the HR-ToF-AMS was switched to sample from the UH/LARGE inlet. No significant changes to species concentrations were detected when switching between the two inlets. Composition from raw HR-ToF-AMS spectra

acquired and averaged over 46 s intervals every 60 s, referred to as 1-minute averages, are reported as well and used to minimize noise at low aerosol mass concentrations typical of the remote free troposphere (FT).

A cloud, aerosol, and precipitation spectrometer (CAPS, Droplet Measurement Technologies, Longmont, Colorado, U.S.A.; Baumgardner et al., 2001) was operated from an underwing pylon by the University of Vienna. The CAPS probe is composed of three instruments: a cloud and aerosol spectrometer (CAS), which measures the concentration of aerosol particles and cloud

droplets from 0.5-50 µm in diameter; a cloud imaging probe (CIP), which measures hydrometeor concentrations from 15 to 930 µm; and a hot-wire cloud liquid water content sensor, which provides a measurement of total liquid water mass concentration in liquid and mixed-phase clouds. The CAPS probe measures aerosol and cloud particle size distributions in situ without substantial heating and drying of the sample stream as is the case for all the other aerosol instruments operating within the fuselage. However, pressure at the location of measurement below the wing is up to 100 hPa higher than ambient (Spanu

et al., 2019), which implies some heating and potential water loss from particles measured by the CAS. The CAPS data are used here to evaluate the performance of the LAS, to examine the effects of efflorescence and inlet losses on the dry size distributions, and to identify cloudy periods for data filtering.



A custom-built, 2-channel cavity ringdown aerosol extinction spectrometer and a 2-channel photoacoustic aerosol absorption spectrometer (together SOAP: spectrometers for optical aerosol properties) replaced the thermodenuded UHSAS during ATom-4.

SOAP measured dry aerosol extinction with an accuracy of 5 % + 0.5 Mm$^{-1}$ at 1 s time resolution behind an impactor with a 50 % efficiency at an aerodynamic diameter of 2.1 μm for a unit density sphere. Finally, aerosol filters and a mist-chamber sampler with on-line ion chromatography were used to measure bulk aerosol composition of the fine and coarse mode (Dibb et al., 1999; Scheuer et al., 2003). The filter samples were analyzed for ionic species by ion chromatography, and for water soluble organic carbon and water- and methanol-soluble brown carbon (Liu et al., 2013). Because the filters measure the total aerosol, and only sulfate and nitrate components are measured in the fine mode, the filter and mist-chamber measurements cannot be compared directly with the size distributions and are not considered further.

## 3 Sampling

The NMASS, UHSAS, LAS, POPS, PALMS and SP2 instruments sampled continuously from the UH/LARGE inlet, a shrouded inlet with a conical inner diffuser with a tip diameter of 6.35 mm (Fig. 3). Total volumetric flow through the inlet varied from ~260 to ~450 volumetric liters per minute (vlpm; 10th and 90th percentiles, respectively). The passing efficiency for this inlet has been quantified as a function of particle diameter by flying the NASA DC-8 aircraft at the same altitude as an instrumented airport control tower located at an elevation of ~800m above sea level (McNaughton et al., 2007). A low altitudes, particles are transmitted through the inlet and plenum system with unit efficiency up to supermicron sizes, after which transmission falls to ~50 % at an aerodynamic diameter of ~5 μm. At an altitude of 12 km, calculations indicate that particles with aerodynamic diameters up to 3.2 μm should have been sampled with >50 % efficiency. In Sect. 6.4 we present in-flight data that are consistent with these transmission efficiencies.

The sample flow exited the UH/LARGE inlet into a 12-cm-long, tapering, stainless steel plenum with a maximum inner diameter (ID) of 5.1 cm and a minimum ID of 1.7 cm. Sampling lines with IDs of 0.76, 1.1, or 1.7 cm extracted air flush with the wall of this plenum (Fig. 3). The NMASS and UHSAS instruments sampled using a 0.76 cm line, which immediately entered a valve and was then reduced to an ID of 0.48 cm. Conductive silicone tubing with an ID of 0.48 cm was used between the valve and a diffusion dryer, and 0.48 cm ID stainless tubing from the dryer to the instruments. Stainless steel ball or plug valves of matching ID were used to isolate each sampling line as needed for calibrations and zero checks.

Excess air from the inlet was exhausted through a 1.7 cm ID port to an exhaust venturi mounted on the side of the aircraft fuselage at the aft end of the passenger cabin. Approximately 90 % of isokinetic flow at the tip of the UH/LARGE inlet was maintained by measuring the mass flow on this line and automatically throttling a butterfly valve based on aircraft true airspeed and the total flow exiting the inlet during ATom. Differences between true airspeed and air velocity at the inlet sampling location outside the aircraft boundary layer forward of the wing leading edge were not accounted for but were a small fraction of true airspeed (McNaughton et al., 2007). A failure of the flowmeter during ATom-3 led to manual manipulation of the butterfly valve. For those affected flights, a parametric relationship between valve setting and flow rate was used to estimate the inlet tip velocity. For all the deployments, a second-by-second, diameter-dependent anisokinetic sampling efficiency factor (Belyaev and Levin, 1974) was applied to the coarse-mode portion of the size distribution to account for the modestly sub-isokinetic sampling. This sampling efficiency was positive (sub-isokinetic flow); concentration enhancements were <20 % for 5 μm particles and <10 % for 1 μm particles (Fig. 4). The sampling efficiency depends on airspeed and is different for ascents, descents and level legs because airspeed differs.

Air sampled by the two UHSAS and NMASS instruments passed through a single-tube Nafion dryer (Model MD-700-12-F3, Perma Pure LLC, Lakewood, New Jersey, U.S.A.). The dried laminar sample flow was carried within stainless steel tubing in a primary sample line. Each NMASS instrument and each UHSAS instruments sampled from this line using tees (Fig. 3),



while one vlpm of excess flow was carried in the line past these tees to a relative humidity (RH) and temperature probe (Model HMP110, Vaisala Inc, Vantaa, Finland), through a volumetric flow controller, and to the exhaust line. The flow rate through the portion of the sample line upstream each of the NMASS tees varied depending on which of the two NMASS orifices was chosen. At sample pressures >425 hPa, a small orifice with a flow rate of 2.1 vlpm was used in each NMASS, and at sample

pressures <425 hPa, a large orifice with a flow rate of 4.0 vlpm was used. Data from each NMASS and UHSAS in the 25 s following the orifice switch were discarded due to resulting sample pressure and flow perturbations. The volumetric flow to the remaining instruments measuring off the UH/LARGE inlet did not vary as a function of altitude. During ATom-4 the SOAP instrument replaced the thermodenuded UHSAS and sampled at 1.0 vlpm.

A second sampling line passed through a separate identical single-tube, vertically oriented Nafion dryer and was immediately

split using "Y" junctions to provide flow to the LAS, POPS, and SP2 instruments. An excess flow of 1 vlpm was carried to a second Vaisala probe, then to a volume flow controller and to the exhaust line. Losses of particles in tubing bends and due to gravitational deposition between the inlet and the LAS instrument (Brockmann, 2001) were calculated on a second-by-second basis and corrections for losses of <30 % for 1 μm particles and <80 % for 5 μm particles (Fig. 4) were applied to the LAS measurements.

The underwing CAPS does not have an active controlled sample flow, but is passively pumped by airflow through the probe. Therefore, its sample flow increases with altitude from ~1 up to ~3.5 vlpm depending on true airspeed (Spanu et al., 2019).

## 4 Data processing

### 4.1 Data selection

Data were reported from aircraft take-off to landing for each ATom science flight at 1 s intervals, with time averaging up to

60 s as described in Sect. 4.3.1. Potential users of the data should be cautioned that the dataset may include periods that are affected from local pollution, biomass burning plumes, air traffic corridors, and Saharan dust layers. The instrument principal investigators listed in the metadata section of the archived files can advise on appropriate screening algorithms. Data from the CAPS probe are used to determine when the aircraft was flying in liquid, mixed-phase, or ice cloud (Dollner et al., in preparation). There is a strong potential for contamination of the sampled aerosol by droplets or ice crystals that shatter on the

inlet tip or within the inlet and produce artifact particles (Weber et al., 1998; Murphy et al., 2004). Size distribution data during all periods within clouds have been removed from the archived dataset.

A goal of the ATom aerosol observations is to combine data from the various particle sizing instruments sampling the dry aerosol downstream of the UH/LARGE inlet into a single size distribution data product, with established uncertainties, that can be used to constrain model simulations of aerosol transport, formation, growth, and removal. To this end the measurements

from the two NMASS instruments and the non-thermodenuded UHSAS and the LAS are merged into a single size distribution. This process involves performing a numerical inversion on the NMASS integral measurements to generate a differential size distribution, interpolating the UHSAS and LAS data into fixed size bins, and averaging as needed to provide a statistically robust size distribution. The merged size distribution does not include larger particles measured by the CAPS probe because of the difference in sample humidity.

### 4.2 Inversion

Each CPC channel, $i$, of the NMASS instruments measures the integral concentration, $\chi_i$, of particles larger than a certain $D_p$. A differential particle size distribution function (e.g., $n(D_p)=dN/dlog_{10}(D_p)$), where $dN$ is the concentration of particles within logarithmic size interval $dlog_{10}(D_p)$), could be determined by differencing adjacent channels. However, the response function, $K_i$, of each CPC channel is not a step function, but rather sigmoidal (Fig. 2; Williamson et al, 2018), so that






$$\chi_i = \int_{3\,nm}^{1000\,nm} K_i(D_p)n(D_p)dlog_{10}(D_p). \tag{1}$$

The response functions slope due to diffusion losses in the sample lines (which change with pressure), diffusion losses internal to the instruments (which do not vary), inherently imperfect channel responses due to spatially inhomogeneous vapor

supersaturations in the condenser, and other effects. To account for these sigmoidal response curves, and to produce a smooth differential particle size distribution with a larger number of logarithmic size bins, $dlog_{10}(D_p)$, than there are measurements, $i$, a non-linear inversion technique is used (Brock et al., 2000; Markowski, 1987; Williamson et al., 2018). This iterative inversion algorithm solves Eq. (1) for a smooth $n(D_p)$ that is consistent with the measured $\chi_i$ within experimental uncertainties. The cumulative concentration reported by the non-thermodenuded UHSAS is used as an additional channel for the inversion. Four

further pseudo-CPC channels, $K_i$, are created by calculating cumulative UHSAS concentrations of particles larger than a given $D_p$. These four channels sum particles larger than 60, 84, 112, and 168 nm, respectively (Fig. 2). These added data provide constraints to the inversion algorithm, forcing the shape of the inverted size distribution at the upper end of the NMASS size range to be consistent with the lower end of the UHSAS size range.

Prior to inversion, the response functions for each of the channels are adjusted to account for size-dependent diffusion losses

in the tubing to the instruments. Internal diffusion losses in the NMASSes are fixed and are inherent in the calibrated response functions of the instruments (Fig. 2). Diffusion losses in the tubing leading to each instrument, however, vary with inlet pressure and also depend on which orifice (thus flow rate) is selected for each NMASS. The diffusion losses are calculated for each second, using measured inlet pressure and calculated sample flows, based on Gormley and Kennedy's (1949) analytical solution for fully developed laminar flow in a pipe of circular cross section (Fig. 5). We do not account for additional

diffusional losses caused by plumbing junctions and tubing bends.

The inversion algorithm generates a size distribution with 20 fixed, logarithmically spaced diameter bins per decade (so $dlog_{10}(D_p)$=0.05) of particle diameter starting at 2.7 nm and extending to 0.3 μm (a total of 42 bins). The UHSAS and LAS data are also interpolated to logarithmically spaced diameter bins with 20 bins per decade extending from 0.067 to 4.8 μm diameter. This interpolation allows constant size bins to be used over the course of all the ATom measurements, even if

calibration of the UHSAS or LAS changes with time. This interpolated size distribution replaces the inverted size distribution for $D_p \geq$ 0.067 μm in the final combined size distribution. A locally-weighted polynomial least-squares (LOESS) regression with an 11-point window is used in the 3 size bins surrounding 0.067 μm to smooth the transition between the inverted and directly measured size distributions. Example cases graphically showing the raw data and the inverted, interpolated and smoothed product are provided in Fig. 6, which also clearly illustrates the rapid time variations in aerosol concentrations that

demonstrate why fast-response size distribution measurements are essential for airborne sampling.

The final interpolated size distribution does not account for systematic differences between the ammonium sulfate optical equivalent diameter from the UHSAS and LAS measurements and the Kelvin diameter from the NMASS measurements. Kupc et al. (2018) found that the geometric diameter of particles with real refractive indices ranging from 1.44 to 1.58 would be reported by the UHSAS with errors of +4 to -10 % respectively, relative to the nominal ammonium sulfate refractive index

(1.53). We have not detected any systematic biases that are dependent on particle composition during calibrations of the NMASS using several different calibrant species (Brock et al., 2000; Williamson et al., 2018). Thus while the UHSAS portion of the size distribution may shift up to 10 % in diameter with to a reasonable variation in refractive index, the NMASS portion of the size distribution will not. The uncertainties produced as a result of this potential mismatch are evaluated in Sect. 5.

Some comments regarding the inversion of NMASS data may help the user understand the limitations of the data. The inversion

between the smallest and next-smallest CPC channels (i.e., between NMASS-1 CPC #1 and NMASS-2 CPC #1; see Fig. 2) is essentially unconstrained. For example a monodisperse aerosol consisting of particles with a diameter of 3 nm would be detected by NMASS-1 CPC #1 with ~30 % efficiency, while another monodisperse aerosol with a diameter of 4 nm would be detected with ~75 % efficiency. Neither aerosol would be detected in the next-largest NMASS channel. There would be no





way to distinguish between particles of these two sizes, and the number of particles produced by the inversion could vary by more than a factor of 2 and still be consistent with the counts in the smallest channel. The way that the inversion handles particles at this end of the size distribution can depend upon the amount of smoothing chosen and even on the initial guess (assigned a flat distribution with $n(D_p) = 1$ cm$^{-3}$ for the ATom dataset). Examining Fig. 6, panels (b2) and (b3), the smallest

particles do not tend toward zero even though there is little evidence in the raw CPC data of a small particle mode. Because of this sizing ambiguity, the NMASS instrument is not appropriate for detailed investigation of initial nucleation and growth mechanisms associated with new particle formation (NPF). For the ATom measurements, we accepted this limitation for several reasons: 1) we could not directly investigate the temporal changes in particle diameter associated with nucleation and growth; 2) there were no measurements of <2 nm particles, which are essential to quantifying nucleation rates; 3) the ATom

mission payload did not include instruments to identify the clusters involved in nucleation (Riccobono et al., 2014); and 4) the probability of making statistically representative measurements of ongoing nucleation events was not high. Because of these limitations, the ATom aerosol measurements were aimed instead at identifying air that was recently influenced by NPF events, mapping out the spatial and thermodynamic patterns associated with high concentrations of recently formed particles, and constraining box and global model simulations with different nucleation mechanisms that produce different geo-spatial

patterns (Kupc et al., in preparation; Williamson et al., submitted manuscript, 2019). We also sought to map the abundance and distribution of particles large enough to serve as CCN and compare these observations with model simulations.

### 4.3 Data products

All quality-controlled final data produced during the ATom project are publicly available (Wofsy et al., 2018). Data derived from the size distribution instruments described here are comprised of three primary products: 1) particle size distributions

reported at 1 s resolution for the nucleation and Aitken modes ($D_p$<0.06 μm), and at variable time resolution for the accumulation mode ($0.06 \geq D_p < 0.5$ μm) and coarse mode $0.5 \geq D_p \leq 4.8$ μm); 2) number, surface, and volume concentrations calculated from these particle size distributions for each of these four modes; and 3) derived products that incorporate measurements from other instruments and/or modeling.

All extensive aerosol properties are reported at standard temperature and pressure (STP), defined here as 273.15 K and 1013

hPa. A conversion factor between the reported STP values and ambient values is provided in the data set (Wofsy et al., 2018); this factor does not account for possible aerosol hygroscopic growth due to an ambient RH that is higher than the measurement RH (usually <20 %). All diameters quoted in this paper are volume equivalent diameters assuming spherical particles (geometric diameters) unless stated otherwise. Care should be taken when comparing to results of other studies as other equivalent diameters are widely used in the literature; see DeCarlo et al. (2004) for further details.

### 4.3.1 Size distributions

The particle size distribution over the size range from 2.7 nm to 4.8 μm is the primary data product from which other parameters are derived. The size distributions between from 2.7 nm $\leq D_p \leq$ 500 nm are produced by combining the NMASS and UHSAS measurements as described in Sect. 4.2. The interpolated LAS size distributions are used for $0.50 < D_p \leq 4.8$ μm. There is no averaging or smoothing at the point where the LAS and UHSAS size distributions merge.

Temporal averaging is used to reduce noise in the final combined size distribution. The size distribution from 2.7-67 nm (the inverted portion of the size distribution) is processed on a 1-s time resolution. Because the UHSAS and LAS have flow rates <1 cm$^{-3}$ s$^{-1}$ and measure larger particles that may be present at low concentrations, the portion of the size distribution with $D_p$ >67 nm is arithmetically averaged with a variable time window to reduce statistical (Poisson) noise. The UHSAS data are summed for $n$ seconds until the total number of particles counted and sized by the instrument reaches 500. The average size

distribution over the $n$-second averaging interval is then calculated. This $n$-second average UHSAS size distribution is then applied to all 1-second size distributions reported within the $n$-second interval. If $n$ reaches 60s, an average size distribution is



calculated and applied regardless of the number of counts reached. The same approach is used for the LAS; however, the LAS portion of the size distribution ($0.5 < D_p \leq 4.8$ μm) is summed until 20 particle counts have been sampled, or until 60 s have elapsed. The values of 500 and 20 counts for the UHSAS and LAS, respectively, were chosen based on observations in the MBL, and result in random fluctuations in reported number concentration due to counting statistics of <25 %.

Because of this variable-interval averaging, the final combined 1 s size distribution may be composed of $dN/dlog_{10}(D_p)$ values that change every second for all $D_p < 67$ nm, while the size distribution of larger particles may change at slower time scales. The advantage of this variable-interval averaging (as opposed to fixed temporal averaging for all instruments) is that fast time response size distributions will be reported at locations where accumulation-mode and coarse-mode particles are abundant, while time-averaged size distributions will be reported for times and diameter ranges where the counting statistics do not

support 1-s measurements. Further averaging beyond 60 s may be needed for the LAS portion of the size distribution ($D_p > 0.5$ μm) in very clean conditions where particle count rates may not reach 20 in a 60 s interval. However, given typical ascent/descent rates of ~500 m/min, averaging beyond one minute risks conflating counting statistics with spatial changes in coarse particle concentrations (Fig. 6).

### 4.3.2 Integral parameters

Weighted integrals of the particle size distributions are used to calculate number (N), surface area (S), and volume (V) concentrations over several size ranges. The nucleation mode is defined as extending from $2.7 \leq D_p < 12$ nm. This definition allows direct comparison with a number of observations (e.g., Clarke and Kapustin, 2002) that use two CPCs with detection diameters of ~3 and ~12 nm to estimate the concentration of "ultrafine" particles. We define the Aitken mode as extending from $12 \leq D_p < 60$ nm (0.06 μm). Particles with $0.06 \leq D_p < 0.50$ μm are associated with the accumulation mode, and

approximate the concentration of CCN for typical liquid water supersaturations. In continental settings it is common to use 1.0 or 2.5 μm as the upper limit for the accumulation mode. Our choice of 0.50 μm is based on the statistics of particle composition reported by the PALMS instrument. These measurements showed that non-volatile sea-salt and dust particles dominated the number and mass concentrations for particles with $D_p \geq 0.50$ μm (Murphy et al., 2019; Froyd et al., manuscript in preparation, 2019). Defining 0.50 μm as the upper limit of the accumulation mode effectively separates particles that are mostly secondary

in origin (the nucleation, Aitken, and accumulation modes) from the coarse mode that contains mostly primary particles in the ATom dataset.

In addition to integral size distribution parameters, we calculated other parameters that use the size distributions along with information from other instruments.

### 4.3.3 Dry submicron aerosol mass

The third moment of the particle number size distribution can be integrated to calculate a total dry aerosol volume. If the density of the aerosol, which may be size-dependent, is known, an aerosol mass can be calculated. A mode-dependent density is estimated as follows: 1) For the nucleation, Aitken, and accumulation modes, we use the non-refractory particle density calculated from the HR-ToF-AMS data. The AMS density is reported at 60 s resolution, so we apply this density to all data within that 60 s interval. 2) For the rBC component, we assume a density of 1.8 kg m$^{-3}$ (Park et al., 2004). 3) For the coarse

mode within the MBL we use the density of dry sea-salt (~2.2 kg m$^{-3}$; Lewis and Schwartz, 2013), but reduce it to 1.9 kg m$^{-3}$ to account for residual water estimated from the PALMS measurements. PALMS derives a density of ~1.45 kg m$^{-3}$ for sea-salt without a sample line dryer, indicating that the particles have not fully effloresced. However, because the UHSAS and LAS sample from actively dried (RH<40 %; often much lower) sampling lines, more complete efflorescence of sea-salt particles is expected. Previous unpublished PALMS observations behind a thermal denuder suggest a sea-salt density of ~1.8-

1.9 kg m$^{-3}$, presumably due to hydrates that resist drying. Based on these arguments, we apply an estimated sea-salt density of 1.9 kg m$^{-3}$. For the coarse mode above the MBL, we assume that soil dust is the primary component, and use a density of 2.6



kg m⁻³ (Wagner et al., 2009). In Sect. 6.5 we detail direct comparisons between the dry mass directly determined by the combined AMS and SP2 measurements and that calculated by integrating over the size distribution measurements.

### 4.3.4 Dry scattering

Scattering at the RH of the measurement (usually <20 %) at STP conditions was calculated directly from the measured size distributions. Because most scattering and extinction is produced by accumulation and coarse mode particles, the UHSAS and LAS measurements contribute most to these parameter. Both the UHSAS and LAS report size distributions that are optically determined assuming a composition of ammonium sulfate with a real refractive index of ~1.52 and no absorbing component. Aged organic aerosol components are believed to have a comparable refractive index (Aldhaif et al., 2018). The same real refractive index is used to calculate the scattering cross-section at wavelength $\lambda$, $Q_{scat}(D_p,\lambda)$, for each particle diameter using
Mie theory, which assumes homogeneous spheres. The total dry scattering $\sigma_{scat}$ is calculated as

$$\sigma_{scat} = \int_{2.7\,nm}^{4.8\,\mu m} Q_{scat}(D_p,\lambda)n(D_p)dlog_{10}D_p. \tag{1}$$

Note that this calculated scattering is at dry, STP conditions, rather than ambient, and does not account for coarse-mode
particles with $D_p$>4.8 μm, which may contribute substantially to total scattering in the MBL and in dust plumes. For the ATom dataset, all optical parameters are calculated for common Aeronet wavelengths of 340, 380, 440, 500, 675, 870, 937, 1020, and 1640 nm, as well as for typical lidar wavelengths of 532 and 1064 nm.

### 4.3.5 Dry asymmetry parameter

Asymmetry parameter, $g$, is the cosine-weighted integrated angular scattering phase function, $P(\theta)$:

$$g = \frac{1}{2}\int_{2.7\,nm}^{4.8\,\mu m}\cos\theta\,P(\theta)\sin\theta\,d\theta. \tag{2}$$

The asymmetry parameter is used in radiative transfer calculations in large-scale models to calculate directional scattering through parameterizations such as the Henyey-Greenstein (Henyey and Greenstein 1941) and delta-Eddington (Joseph et al.,
1976) approximations. The value of $g$ is calculated for the same wavelengths as was used in the calculation of dry scattering, does not include the contribution from particles with $D_P$>4.8 μm, and does not represent conditions at ambient RH.

### 5 Biases and uncertainties

Size distributions determined by combining measurements from multiple instruments using different techniques are subject to different types of errors. Systematic errors are caused by losses or enhancements of particles during sampling and transport,
mis-sizing of particles due to particle refractive index or shape, calibration biases, and flow, pressure, and temperature calibration errors (the last two affecting the STP correction). Some of these potential biases may be size dependent. Additional systematic biases may be caused to different definitions of particle diameter. For example, the NMASS measures a Kelvin-equivalent diameter (closely related to geometric size), while the UHSAS measures an optical-equivalent diameter. Random uncertainties include those due to measurement repeatability, calibration, flow, pressure and temperature repeatability, and
counting statistics. Systematic and random uncertainties may combine in unexpected ways in the extensive processing performed to combine the 10 CPCs within the NMASS instruments with the UHSAS and LAS size distributions.

Uncertainties in the different reported size ranges are summarized in Table 1 and detailed below. Uncertainties associated with the NMASS instruments dominate the nucleation and Aitken mode uncertainties, and have been calculated for a range of cases (Williamson et al. 2018). These uncertainties were determined by Monte Carlo simulation of measurement and calibration



uncertainties propagated through the numerical inversion code for eight representative size distributions. Uncertainties in integrated number, surface, and volume were <20 % except for the case of a low concentrations of particles near the bottom end of the detection range of the instrument, which were as high as 39 %. As discussed in Sect. 4.2, there is ambiguity in determining the number of particles with $D_p$ <4.5 nm because only one NMASS channel detects these particles. Also as

discussed in Sect. 4.2 the NMASS measurements are corrected for diffusion losses, which vary with flow rate and pressure (Fig. 5) and are calculated on a second-by-second basis.

Uncertainties in the accumulation mode are dominated by uncertainties in the UHSAS instrument. As discussed in Kupc et al. (2018), potential calibration, flow, and pressure biases produce uncertainties in particle number, surface, and volume concentrations of ±3.9 %, +8.4/-17.8 % and +12.4/-27.5 %, respectively. Counting statistics can introduce a large random

uncertainty because of the relatively low sample flow rates of the UHSAS and LAS (~1 cm$^3$ s$^{-1}$). At extremely low concentrations sometimes encountered (<<1 % of the data set), these uncertainties can exceed 100 % for 1-second data. For more typical concentrations, and averaging over 10 seconds, uncertainties due to counting statistics in the accumulation mode were often <15 %.

Systematic uncertainties in the coarse mode are dominated by uncertainties in the LAS instrument and in assumptions about

particle shape and refractive index. As with the UHSAS, potential sizing biases due to lack of knowledge of particle refractive index can produce uncertainties in particle surface and volume of ~30 %. However, outside of the MBL, the coarse mode is often dominated by aspherical particles composed of dust, fly ash, sea-salt, etc., with a range of refractive indices (e.g., Petzold et al., 2009; Weinzierl et al., 2011; Froyd et al., manuscript in preparation, 2019). Because of this, it is difficult to assign an estimate of bias based on first principles. Based on size calibrations with different aerosol types, flow-meter calibrations, and

comparisons with the UHSAS (Sect. 6.2), we conservatively estimate potential coarse-mode number, surface, and volume systematic uncertainties of <5 %, <25 %, and <50 %, respectively, for the coarse-mode aerosol during ATom-1. Random uncertainties were often the dominant source of uncertainty for the low-concentration coarse mode. Averaging to 20 counts in the LAS (Sect. 4.3.1) will produce an uncertainty in coarse particle number of 22 %. However, averaging of the size distribution is not permitted to exceed 60 s, which may result in many fewer counts and in larger random uncertainties. Random

uncertainties in coarse surface area and volume were estimated for the MBL and for the FT by examining the geometric variation of these parameters during several extended (>5 min) periods of level flight when accumulation mode aerosol parameters showed little variation. The random variation of 60-s geometric means for both surface area and volume in the FT exceeded (+100 %, -49 %). The user of the coarse-mode aerosol data is encouraged to increase the time averaging to achieve counting statistics that allow quantitative use as needed.

Coarse particles measured by the LAS may be subject to inertial or gravitational losses in the sampling lines. Inertial losses are small and are corrected as described in Sect. 3. Possible inertial losses within the turbulent, diffusing inlet and the plenum are not

accounted for. However, in Sect. 6.4 we provide in-flight evidence that losses for coarse mode, dry aerosol particles are small for $D_p$ <4.8 μm at altitudes <3 km and <3μm at altitudes >7 km, roughly consistent with previous inlet testing (McNaughton

et al., 2007).

## 6 Instrument and data comparisons

### 6.1 UHSAS vs. UHSAS

During ATom-1, the second UHSAS instrument was operated without the thermodenuder installed, allowing direct comparison between the two instruments. Kupc et al. (2018) show that the agreement between the instruments was well within stated

experimental uncertainty. Integral number, surface, and volume concentrations for particles with $D_p$ between 0.07 and 1.0 μm agreed with 2-sided linear least-square slopes of 0.99, 0.97, and 0.95 respectively, and Pearson correlation coefficient values





$(r^2)$ >0.98. During ATom-2 the thermodenuder was installed and operated at 300ºC. During brief periods in the MBL when the thermodenuder was bypassed, agreement between the UHSAS instruments was similar to that measured during ATom-1.

### 6.2 UHSAS vs. LAS

The non-thermodenuded UHSAS and the LAS were compared over the region of overlap between the two instruments.
Number and volume size distributions were integrated between 0.104 to 0.97 μm for the two instruments. A scatterplot of 10 s data between the two instruments for all ATom-1 flights shows agreement within 1 % for particle number and 9 % for integrated volume (Fig. 7). The value of $r^2$ is 0.96 for particle number and 0.80 for the volume comparison. The instruments produce different sizing biases for particles with refractive indices or shapes that differ from the ammonium sulfate calibration aerosol, such as Saharan dust (Petzold et al., 2009; Weinzierl et al., 2011), as demonstrated in Fig. 7. Although the UHSAS
and LAS share nearly identical scattering geometries, their lasers are at different wavelengths (1053 nm for the UHSAS and 633 nm for the LAS), which produces different responses to dust aerosol. Outside of the dust, the agreement in particle volume is well within the stated systematic uncertainties of the two instruments (Table 1).

### 6.3 NMASS vs UHSAS

The channel of the NMASS that nucleates and counts the largest particles has a 50 % efficiency $D_p$ of ~59 nm. The UHSAS
without the thermodenuder counts particles 63 nm with ~50 % detection efficiency (Kupc et al., 2018). The shapes of the response curves are different for the two instruments, with the NMASS response being more sloped and sigmoidal, while the UHSAS response rises more rapidly to count with 100 % efficiency for $D_p$>75 nm (Fig. 2). We would expect that the concentrations measured by these two instruments would be correlated, but that the NMASS concentrations would exceed those of the UHSAS, except in cases where most particles were >120 nm where NMASS efficiency falls. As shown in Fig. 8,
the upper NMASS channel and the total UHSAS concentration are correlated as expected for the entire ATom-1 dataset $(r^2$=0.87), while the NMASS concentrations generally exceed the UHSAS concentrations as expected (slope=1.44) except at concentrations <10 cm$^{-3}$ where only accumulation mode particles were present.

### 6.4 LAS vs. CAPS

The CAPS instrument measures at near-ambient conditions from an underwing canister, while the other size distribution
instruments sample within the cabin behind the UH/LARGE inlet, transport plumbing, and a diffusion dryer. Particles within the cabin are subject to heating from air deceleration (ram heating) and convective heat transfer from the sampling lines. Thus the cabin and CAPS measurements are comparable without correcting for aerosol water content only during periods of low humidity. The LAS instrument provides the coarse portion of the aerosol size distribution measurement. Because their optical geometries and wavelengths of the CAPS and LAS are different, they respond differently to particle size, refractive
index, and shape. Nonetheless, comparisons between the instruments provide an important check on instrument performance, and in dry conditions allow a qualitative evaluation of inlet passing efficiency. Figure 9 shows three example size distributions. Two of these were measured in dry conditions; one in Saharan dust at altitudes <3 km (Fig. 9a) and one in low-concentration conditions at altitudes >7 km (Fig. 9b). A third size distribution was measured in the MBL in sea-spray aerosol at a relative humidity of 79 %. Agreement between the LAS and the CAPS probe is within uncertainties for the two
dry cases. At high altitude (Fig. 9b), the size distribution from the cabin instruments reports no particles larger than 3 μm due to inlet losses and poor counting statistics, while the CAPS shows a continuous size distribution to larger sizes. At low altitude (Fig. 9a), the combined size distribution extends to 4.8 μm. These differences are consistent with the expected altitude dependence of inlet losses (McNaughton et al., 2007). At high humidity in the MBL (Fig. 9c), the effect of hydration on the sea-spray aerosol is readily apparent, as the CAPS probe shows considerably more particles than do the cabin





instruments in their region of overlap. Note that in all cases coarse particles at sizes >4.8 µm are present, so the in-cabin measurements provide a lower estimate of the total coarse aerosol.

A more systematic evaluation of the overlap region between the CAPS instrument and the LAS measurements between ~0.7 and 1.80 µm in all dry conditions encountered in ATom-1 shows agreement within expected uncertainty for both number

(Fig. 10a) and volume (Fig. 10b), with regression slopes of 1.24 and 0.96, respectively and $r^2$ values of 0.99. At low concentrations the comparisons are noisy because of poor counting statistics, primarily due to the low flow rate of the LAS.

### 6.5 Mass calculated from size distribution vs AMS mass

The AMS measures the mass concentration of non-refractory aerosol components over a diameter range from ~0.02 to ~0.7 µm geometric diameter following a polygonal (in log diameter space) efficiency curve (Zhang et al., 2004; Knote et al., 2011;

Hu et al., 2017), with particles between ~0.04 and ~0.5 µm geometric diameter measured with >50 % efficiency. For the HR-ToF-AMS, as configured and operated during the first three deployment of the ATom project, the apices of this transmission polygon were at 0.035, 0.10, 0.50, and 1.2 µm vacuum aerodynamic diameter. By integrating the measured size distributions over the AMS size range while accounting for the polygonal efficiency curve, and applying an estimate of particle density, an AMS-equivalent aerosol mass can be calculated. The density of the non-refractory components is estimated from the measured

AMS composition and reported in the AMS file on the data archive. Total mass is estimated by adding rBC from the SP2 measurements over the mass-equivalent diameter range of 0.09-0.55 µm; The rBC mass is typically <2 % of the total mass, and the assumed density of the rBC is 1.8 kg m$^{-3}$.

During ATom-1, the 1-minute average mass calculated as described above and that measured directly by the HR-ToF-AMS and the SP2 agreed within stated uncertainties (Fig. 11), with a slope of 1.04 and an $r^2$ of 0.84 above the MBL and outside of

dust plumes, which are excluded from the comparison because of substantial refractory aerosol fractions.

### 6.6 Scattering calculated from size distribution vs. SOAP extinction

For a non-absorbing aerosol, scattering and extinction are equivalent. The remote aerosol measured during most of ATom-4 (when the SOAP instrument was operated) can be considered to be non-absorbing for the purpose of instrument comparison because rBC was generally a small fraction of the aerosol mass. Aerosol scattering was calculated from the size distributions

at 532 nm assuming the refractive index of ammonium sulfate. Scattering from each size bin was integrated up to the calculated 50 % efficiency of the 2.1 µm aerodynamic diameter of the impactor in front of the SOAP instrument, adjusting for sample pressure and using the mean density reported by the AMS during ATom-4 of 1.67 kg m$^{-3}$. The calculated scattering compared with the SOAP extinction measured at 532 nm had a slope of 0.90 with $r^2$=0.97 for one-minute averaged data (Fig. 12). The agreement between calculated scattering and measured extinction is within the combined instrumental uncertainties.

**7 Summary**

The ATom observations have produced a unique dataset that provides high-resolution snapshots of global-scale remote aerosol properties between ~150 m and ~12 km altitude between the Arctic and the Antarctic over both the Pacific and Atlantic Oceans in four seasons. This dataset will be useful for constraining global model simulations of aerosol abundance and characteristics and for understanding aerosol sources and sinks in the remote troposphere. These observations also provide important

constraints on estimates of pre-industrial aerosol abundance and dry aerosol properties, including CCN concentrations, scattering, and asymmetry parameter.

The ATom size distributions were produced by combining measurements from several instruments operating inside the cabin of the NASA DC-8 as it profiled over both the Pacific and Atlantic Oceans. The size distributions were corrected for inlet sampling (aspiration) efficiency and for diffusional losses, and uncertainties in integrated properties were calculated using



Monte Carlo methods, repeated calibrations, and evaluation of in-flight variations. Comparisons between parameters measured by different instruments show levels of agreement within the calculated uncertainties, except between two optical particle counters in the case of Saharan dust. Particle mass estimated from the integrated size distributions agreed with independently measured non-refractory particle mass measured by an AMS and SP2. Comparisons between the in-cabin measurements and

under-wing probes suggest that the inlet efficiently sampled particles with $D_p$<4.8 µm at low altitudes and <3 µm at high altitudes.

In the future we will use these size distributions, size-resolved compositional information, and measurements of $H_2O$ mixing ratio, temperature, and pressure to estimate aerosol hygroscopicity. By accounting for water mass and combining the in-cabin size distributions with those for coarse-mode particles measured underwing by the CAPS probe, we will calculate ambient

aerosol properties and related parameters. This extended dataset of aerosol characteristics at ambient conditions will improve understanding of the optical properties of the remote and pre-industrial aerosol and will be useful for satellite comparisons, radiative transfer calculations, and model evaluations.

*Author Contribution.* All co-authors made measurements that were used in this manuscript. CAB performed the analyses and

prepared the manuscript with contributions from all co-authors.

*Competing Interests.* The authors declare that they have no conflicts of interest.

*Disclaimer.* The contents do not necessarily represent the official views of NOAA or of the respective granting agencies. The

use or mention of commercial products or services does not represent an endorsement by the authors or by any agency.

*Acknowledgements.* The authors acknowledge support by the U.S. NASA's Earth System Science Pathfinder Program under award NNH15AB12I, NNX15AJ23G and NNX15AH33A, NASA award 80NSSC19K0124, and by the U.S. National Oceanic and Atmospheric Administration (NOAA) Health of the Atmosphere and Atmospheric Chemistry, Carbon Cycle, and Climate

Programs. AK is supported by the Austrian Science Fund FWF's Erwin Schrodinger Fellowship J-3613. BW and MD have received funding from the European Research Council (ERC) under the European Union's Horizon 2020 Research and Innovation Programme under grant agreement No. 640458 (A-LIFE) and from the University of Vienna.

*Data Availability.* The data are available as given in Wofsy et al. (2018), and may also be accessed at

https://espoarchive.nasa.gov/archive/browse/atom.

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





Table 1. Estimated systematic and random uncertainties

| Mode | Diameter range | Number | | Surface | | Volume | |
|---|---|---|---|---|---|---|---|
| | | systematic % | random % | systematic % | random % | systematic % | random % |
| nucleation | 3-12 nm | ±20 | ±13 | ±3 | ±10 | ±5 | ±12 |
| Aitken | 12-60 nm | ±10 | ±6 | ±5 | ±7 | ±2 | ±18 |
| accumulation | 60-500 nm | ±3.9 | typically ±15 (10s average) | +8/-18 | typically ±15 (10s average) | +12/-28 | typically ±15 (10s average) |
| coarse | 500-4800 nm | ±5 | typically ±22 (60 s average) | ±25 | MBL +64/-30 FT +106/-49 (60 s average) | ±50 | MBL +73/-41 FT +144/-55 (60 s average) |

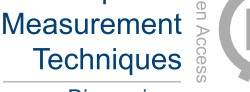

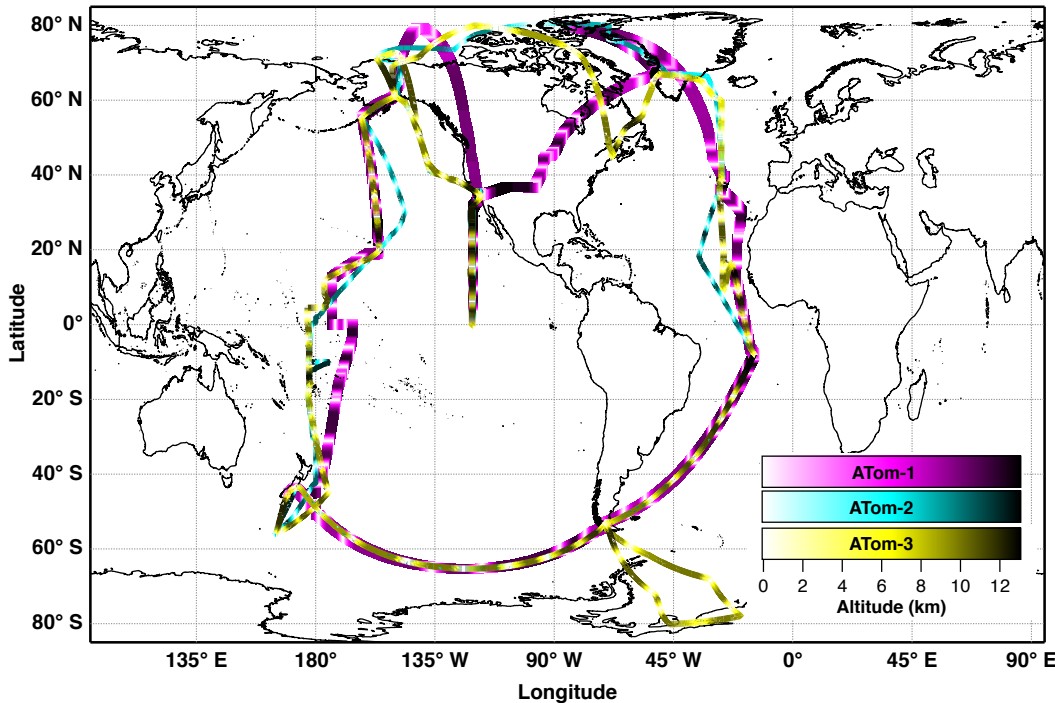

**Figure 1.** Flight tracks of the NASA DC-8 for ATom-1 (magenta), ATom-2 (cyan), and ATom-3 (yellow), shaded to show altitude.





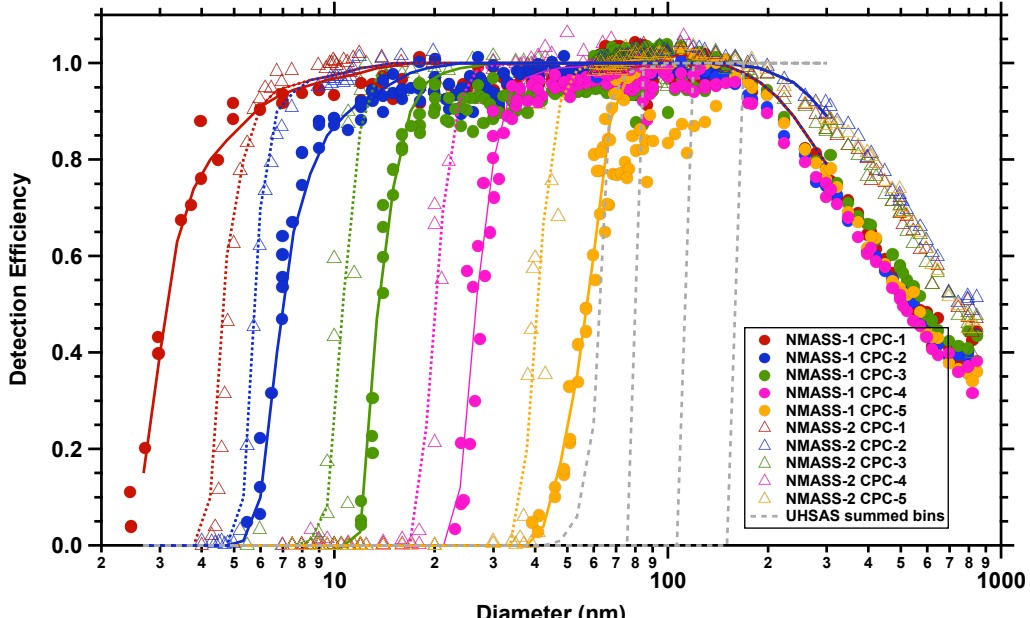

**Figure 2.** Detection efficiencies for the five CPCs in each of two NMASS instruments as operated beginning in ATom-2. Scatter in the points shows repeatability from multiple calibrations. Lines are fits used as $K_i(D_p)$ in the inversion of Eq. (1).

5    Cumulative bins from the UHSAS instrument (dashed grey lines) are used in the inversion of NMASS data to recover a size distribution for diameters from 2.7-300 nm. Decreases in detection efficiency for diameters >150 nm are due to impaction losses in the pressure reduction section, and are different for the two NMASSes. During ATom-1 only the NMASS-1 instrument was operated.





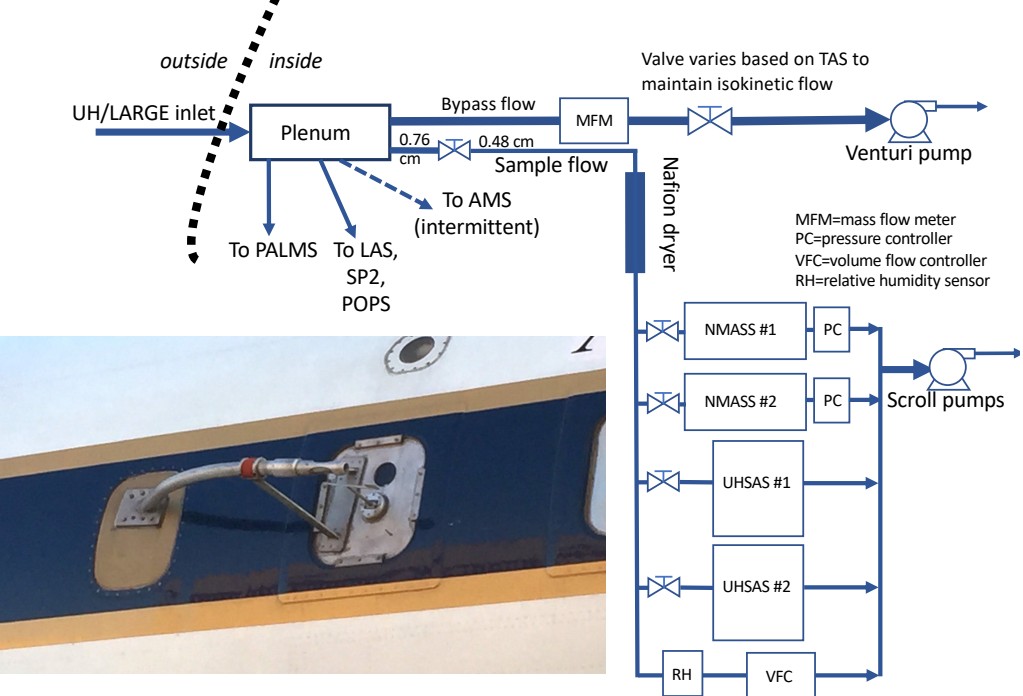

**Figure 3.** Schematic of the sampling system for the aerosol size distribution instruments during ATom. Lower left: Photograph of the shrouded UH/LARGE inlet mounted on the NASA DC-8 fuselage. MFM=mass flowmeter; PC=pressure controller, RH= relative humidity sensor; VFC=volume flow controller.





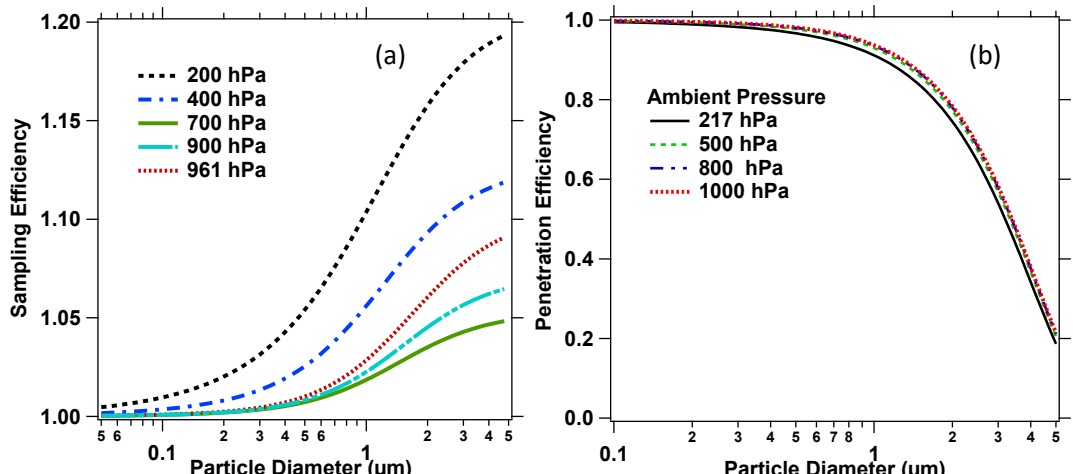

**Figure 4. (a)** Calculated sampling (aspiration) efficiency of the aerosol sampling inlet for five pressures encountered during ascent. The curve for 961 hPa is in level flight at the bottom of a profile. **(b)** Calculated penetration efficiency of particles

5  reaching the LAS instrument through the sample tubing as a function of particle diameter.





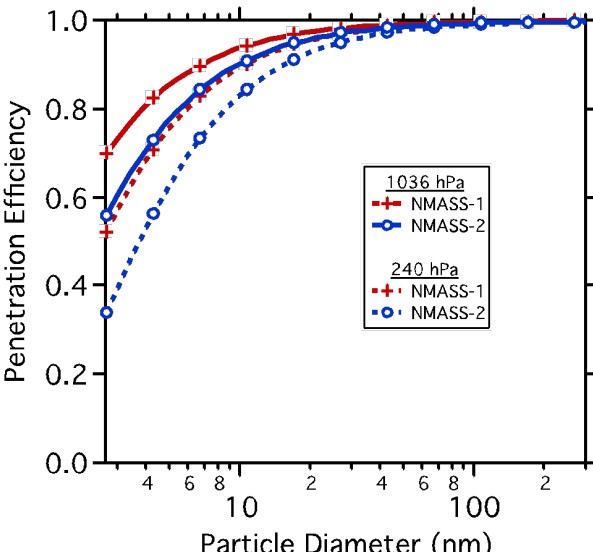

**Figure 5.** Calculated penetration efficiency for diffusing particles in the tubing between the inlet plenum and the NMASS-1 and NMASS-2 instruments for two pressures near sea level (1036 hPa) and near aircraft maximum altitude (240 hPa).

5    Penetration efficiency for NMASS-2 is lower because the tubing between the inlet and the instrument is longer.





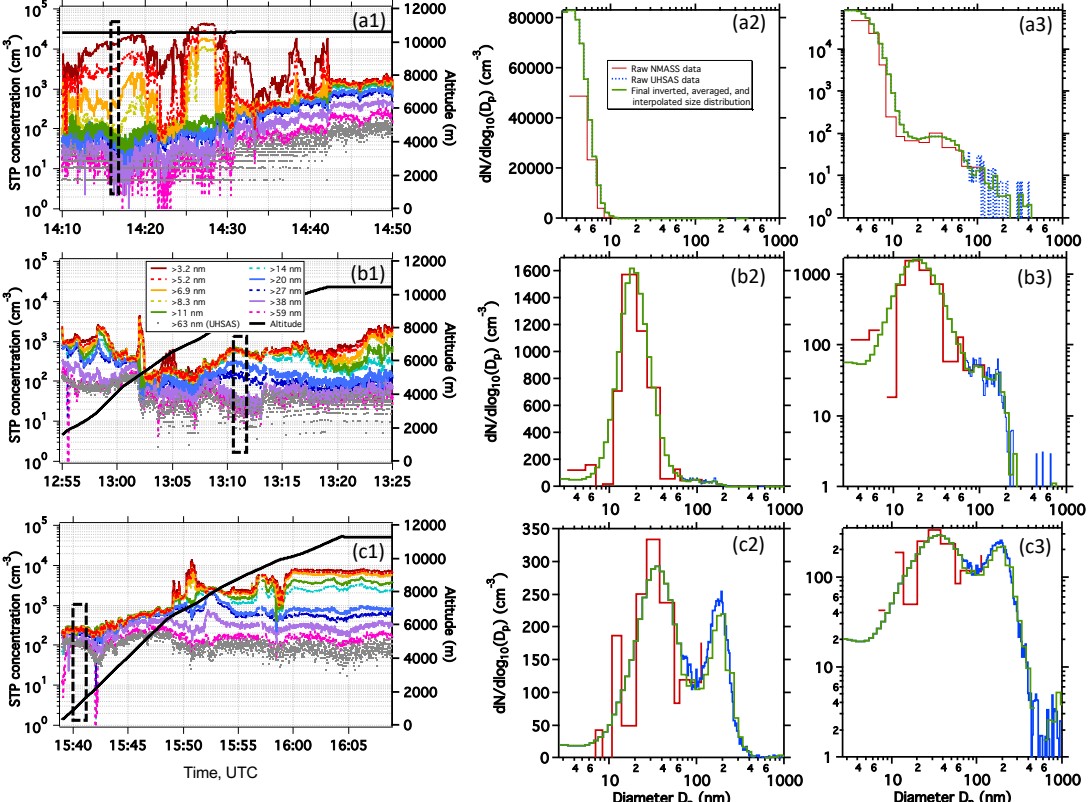

**Figure 6. (a1, b1, c1)** Three example time plots of STP concentrations measured by each channel of the two NMASS instruments and by the non-thermodenuded UHSAS on 13 February 2017 on a flight over the South Atlantic Ocean between
5  Punta Arenas, Chile (43.2° S, 70.9° W), and Ascension Island, UK (7.9° S, 14.3° W). Each plot shows 30 minutes of flight time and uses the same scales. **(a2, b2, c2)** Linear particle number size distributions from average CPC concentrations over the interval shown by the dashed rectangular boxes in (a1), (b1), and (c1), respectively. These size distributions show raw NMASS size distributions (red curve) obtained by simple differencing of adjacent CPC channels using the 50 % efficiency values of each CPC response curve (Fig. 2). The blue curve shows the raw UHSAS size distribution, which is reported in 99
10  channels. The green curve shows the inverted NMASS data and the time-averaged and size-interpolated UHSAS data as reported in the ATom archive using constant logarithmic size bins. **(a3, b3, b3)** The same size distributions plotted using a logarithmic y-axis to show the accumulation mode better.





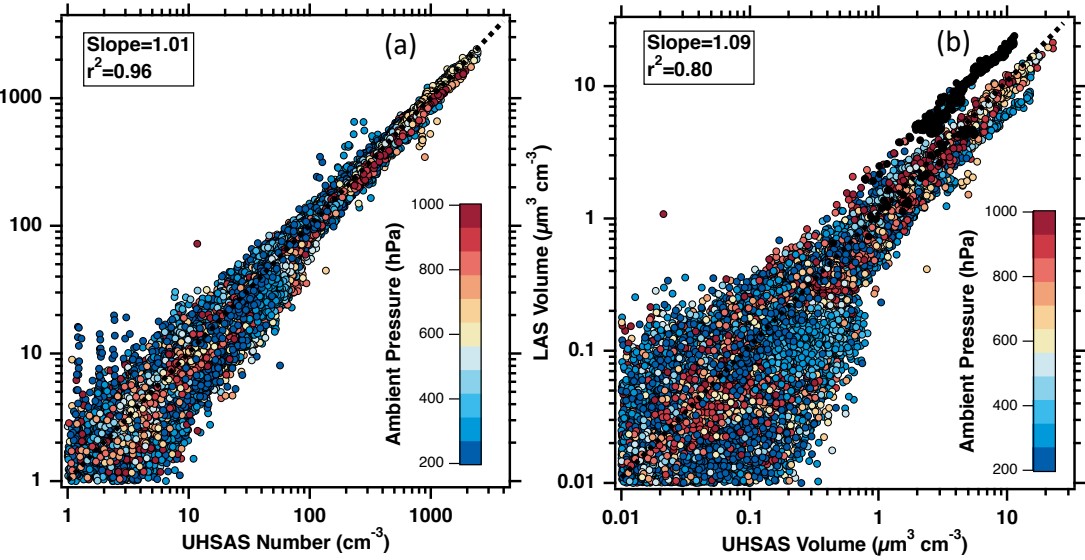

**Figure 7. (a)** Integrated ten-second average particle number concentrations from 0.104 to 0.97 µm diameter from the UHSAS
and LAS instruments for all of the ATom-1 deployment. **(b)** As for (a), but for integrated particle volume. Colors indicate
ambient pressure. Slopes from two-sided least squares regressions forced through a zero intercept are shown. Values of r² are

5      from a one-sided linear least-squares regression. Black points in the volume plot show periods of dust from Africa.





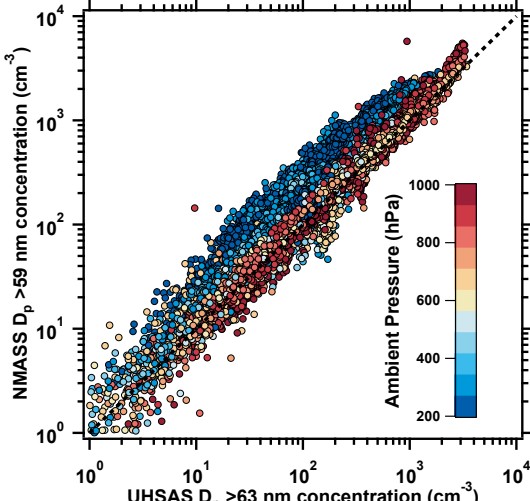

**Figure 8.** Ten-second particle number concentration from the largest NMASS CPC channel as a function of the concentration measured by the non-thermodenuded UHSAS for all of the ATom-1 deployment. Line is the 1:1 ratio. The NMASS concentrations are higher than the UHSAS concentrations at lower pressures because the aerosol size distribution is more steeply sloped toward smaller particles, while within the boundary layer the accumulation mode dominates.



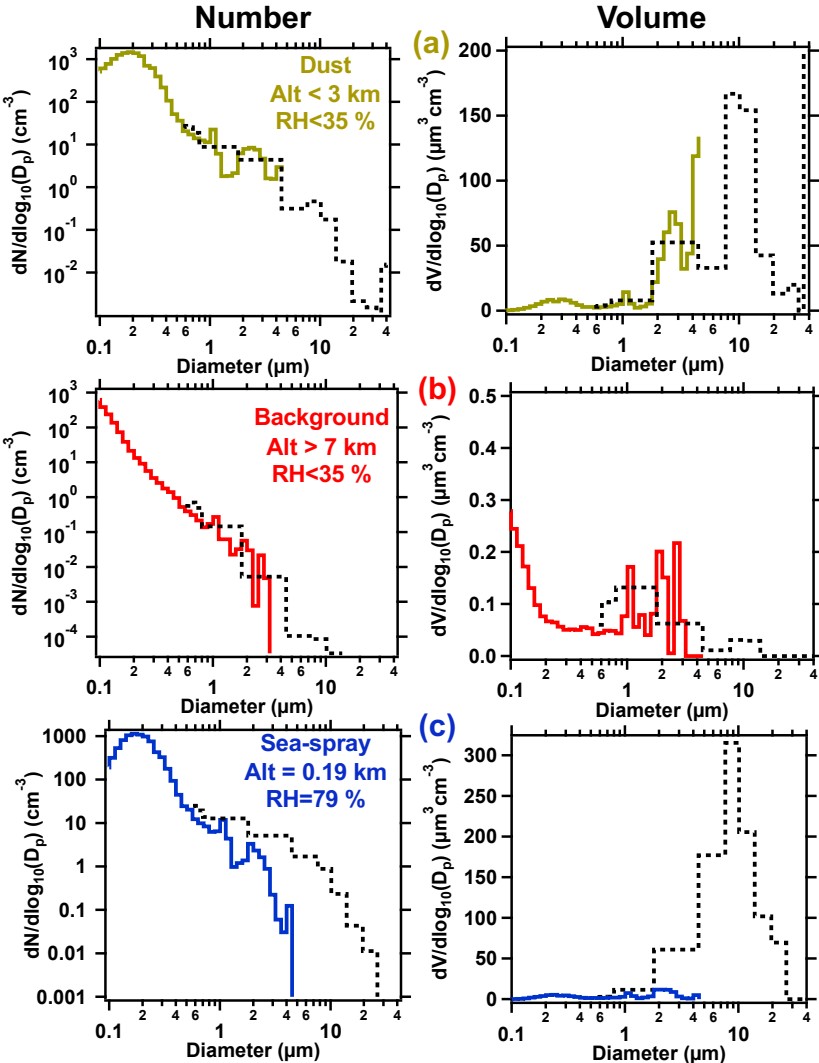

**Figure 9. (a)** Number size distributions shown on a logarithmic scale (left column) and volume size distributions on a linear scale (right column) from measurements in a Saharan dust plume over the Atlantic Ocean (26° N, 20° W) on 2016/08/16 at aircraft altitude <3 km and RH < 35 %. Dry size distributions from the in-cabin instruments are shown with solid lines; near-ambient size distributions from the under-wing CAPS probe are shown in dashed lines. **(b)** As for (a), but measured when altitude >7 km and RH <35 %. **(c)** As (a), but measured for 5 minutes at 190 m in the MBL at *RH*=79 %. Hydrated sea-spray particles comprise the coarse mode, for which >80 % of the particle volume is water.





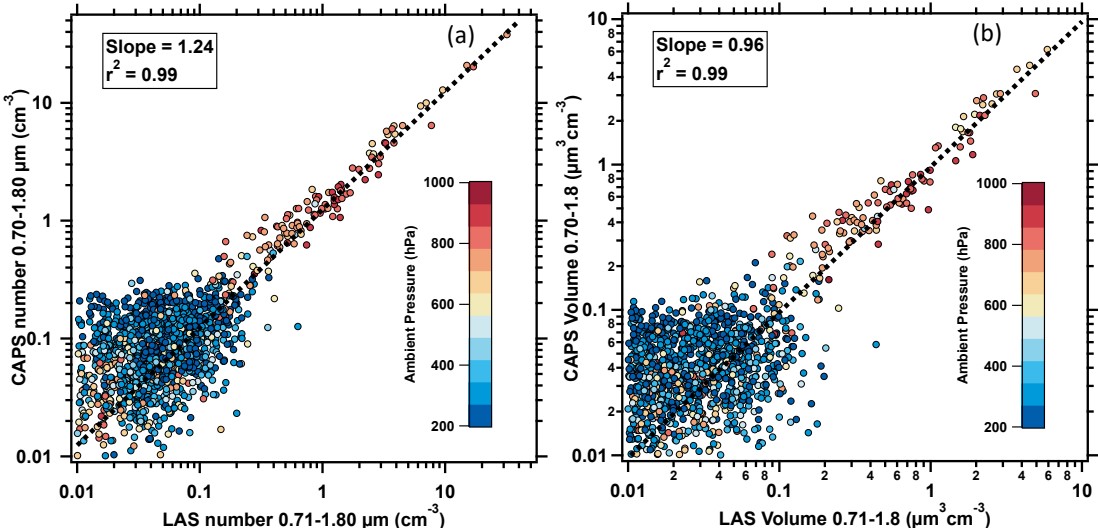

**Figure 10. (a)** One-minute average CAPS number concentrations for diameters from ~0.7-1.80 µm plotted against the same parameter from the LAS instrument for periods of flight during ATom-1 when RH with respect to liquid water was <40 %. **(b)** As in (a), but for particle volume. Lines and slopes are from two-sided least-squares linear regression forced through zero.

5    Values of $r^2$ are from a one-sided linear least-squares regression.





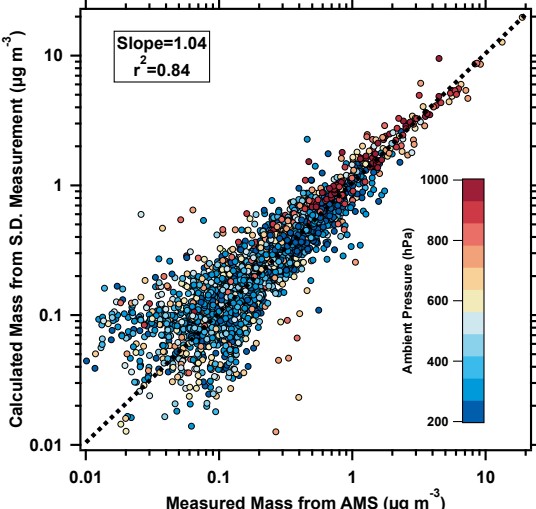

**Figure 11.** Dry aerosol mass calculated by integrating the particle size distributions measured by the NMASS, UHSAS, and LAS instruments and applying the density and AMS aerodynamic lens transmission efficiency calculated as described in the text as a function of the aerosol mass measured by the AMS and SP2 instruments. Data are one-minute averages from the entire ATom-1 campaign. Line and slope are from a two-sided linear least-squares regression forced through zero intercept. Value of $r^2$ is from a one-sided linear least-squares regression.





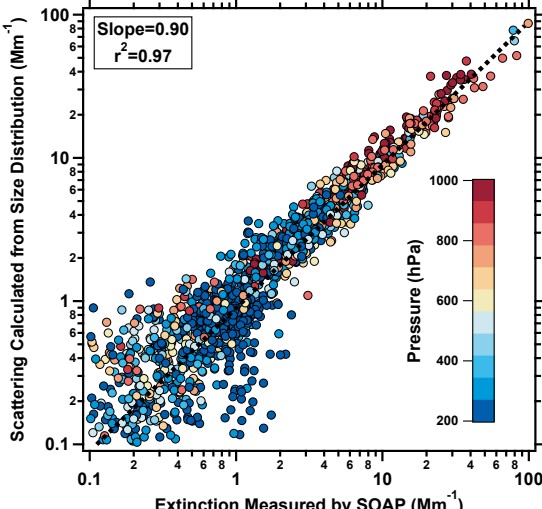

**Figure 12.** Dry aerosol scattering at 532 nm calculated using Mie theory from the particle size distributions measured by the NMASS, UHSAS, and LAS instruments as a function of the dry aerosol extinction at 532 nm measured directly by the SOAP instrument. Data are 60 s averages from the entire ATom-4 campaign. Line and slope are from a two-sided linear least-squares fit forced through zero intercept. Value of $r^2$ is from a one-sided linear least-squares regression.