# Peer review of "Aerosol size distributions during the Atmospheric Tomography"

_Atmospheric Measurement Techniques, 2019_

## Referee Comment (RC1) · Anonymous Referee #1 · 23 Apr 2019

This manuscript describes in detail the aerosol particle measurements carried out during the ATom missions onboard the NASA DC8. It describes the measurement systems and the generated data products which will be freely available to the scientific community. This data set is of high relevance for atmospheric research and will certainly be widely used. The manuscript is very well-written and should be published in AMT after the following minor points have been considered:

page 4 line 6-7: how many size classes can the POPS determine?

page 4 lines 8-15: in which format will PALMS data by made public on the data base?

page 4, line 25: HIMIL and UH/LARGE acronyms are not explained (appear here for

the first time).

page 5, line 35: Please give the equation used for the efficiency factor. I assume it is Equ. 9 in Belyaev and Levin?

page 6 line 12: replace by "...and the LAS instrument were calculated after Brockmann (2001) on a second-by-second..."

page 7, line 30ff / Fig 6: What was the averaging time for the size distributions? It's hard to see from the graphs a1, b1, c1. These are not the reported 1-second data products but rather 30 second avarages.

page 8, line 20: please refer to section 4.3.1 when mentioning that the time resolution is variable.

Section 4.3.1: looking at Fig 6, b2 and c2, the first channels of NMASS (< 15 nm) may also suffer from poor counting statistics. Why is the averaging not applied to the first NMASS channels as well?

page 9 line 18: Any literature references for typical Atiken mode size range definitions?

page 9 line 32: The AMS measures only the accumulation mode. The density of the nucleation mode may be different (higher, if assuming that H2SO4 plays an important role for nucleation).

page 11 line 32: remove line break.

page 12 line 14: this is channel NMASS-1 CPC5 in Figure 2, right?

page 12, line 35-36: That would mean that the inlet transmission is unity up to 3  $\mu$ m and drops immediately to zero. On page 5 it was said that at 12 km the transmission at 3  $\mu$ m is still 50%. Would it be possible to average the >7 km distibution over a longer time? What does "> 7 km" mean exactly? Can you give the altitude range that is averaged here?
page 13, lines 3-6 and Figure 10: wouldn't it make sense to average the high altitude values over more than 1 minute to improve counting statistics?

Figures

Figure 1: What about ATom-4?

Figure 7: Y-axis "LAS number" is missing. In general, shouldn't the axes be called "number concentration" and "volume concentration"? (same for Fig. 10)

Figure 8: Fig 7, Figs 10, and 11 show the regression slope and r2, while Fig. 8 shows the 1:1 line. Wouldn't it be better to make the Figures consistent?

---

## Referee Comment (RC2) · Anonymous Referee #2 · 25 Apr 2019

Review amt-2019-44

GENERAL COMMENT

The manuscript describes with large richness of detail the methods applied for obtaining particle size distributions from the aerosol instrumentation deployed during ATom 1 to 4 on the DC-8, including the uncertainties associated to the parameters and derived data products, such as the aerosol scattering coefficient or the aerosol mass concentration. The applied methods and associated uncertainties are described very carefully, whereas in-depth intercomparisons between different instruments and, wherever appropriate, between instrument parameters and data products demonstrate the

high data quality of the ATom data set. The manuscript will serve as the reference document for any scientific analysis based on the ATom aerosol data set. It is therefore of high relevance for the scientific community and will set a standard for future research campaigns using instrument combinations.

The manuscript fits perfectly into the scope of AMT. It is clearly structured, well organized and very well written. Only technical corrections are required before the manuscript is acceptable for publication in AMT.

MINOR COMMENTS

1. Page 2, line 6: the authors refer sometimes to "the paper", sometimes to "the manuscript". I suggest using "the manuscript" throughout the text.

2. In the abstract, the authors specify the range of particle measurements from 2.7 nm to 4.8 diameter (page 1, line 18), whereas in the instrumentation section they specify the range as from 3 nm to 930 $\mu$m diameter (page 2, line 17). These different ranges are caused by instrument specificities, but later in the manuscript the authors never used the range from 4.8 $\mu$m and larger. It might be worthwhile to state this in the abstract.

3. On page 3, line 9, the authors may add "this refractive index range 'of the atmospheric aerosol' ".

4. On page 4, line 26, there is a typo "located 'aft' of the UH/LARGE inlet." Please correct.

5. On page 5, line 5, the authors introduce filter samples collected during the flight which were used for post-flight chemical analyses. Please state the sampling time and the resulting spatial resolution.

6. On page 6, line 15ff, the authors mention briefly the flow through the CAPS instrument and refer to Spanu et al. (2019) for the flow-induced errors in aerosol. Compared to the detailed discussion of the other instruments' uncertainties, a few more details

would be good. In particular, how does the analysis of corrections published by Spanu et al. (2019) compare to the recently published detailed study on the thermodynamic correction of particle concentrations measured by underwing probes on fast-flying aircraft (Weigel et al., 2016)?

7. On page 11, line 32, there is an erroneous line break inserted.

8. In Fig. 1, neither the track for Atom-4 is shown nor is an explanation given why this is not the case. Please specify.

9. In Fig. 7, the y-axis title is missing for the left panel.

10. In Fig. 10, the title of the y-axis is too close to the axis labels for both panels.

REFERENCES

Spanu, A., Dollner, M., Gasteiger, J., Bui, T. P., and Weinzierl, B.: Flow-induced errors in airborne in-situ measurements of aerosols and clouds, 2019, 1-46, doi: 10.5194/amt-2019-27, 2019.

Weigel, R., Spichtinger, P., Mahnke, C., Klingebiel, M., Afchine, A., Petzold, A., Krämer, M., Costa, A., Molleker, S., Reutter, P., Szakall, M., Port, M., Grulich, L., Jurkat, T., Minikin, A., and Borrmann, S.: Thermodynamic correction of particle concentrations measured by underwing probes on fast-flying aircraft, Atmos. Meas. Tech., 9, 5135-5162, doi: 10.5194/amt-9-5135-2016, 2016.

---

## Author Comment (AC1) · 8 May 2019

We thank both referees for their thorough and helpful reviews of this manuscript. We respond to each comment below in brackets.

Reviewer #1

This manuscript describes in detail the aerosol particle measurements carried out during the ATom missions onboard the NASA DC8. It describes the measurement systems and the generated data products which will be freely available to the scientific community. This data set is of high relevance for atmospheric research and will certainly be

widely used. The manuscript is very well-written and should be published in AMT after the following minor points have been considered:

[Thank you for the positive comments on the manuscript.]

page 5, line 35: Please give the equation used for the efficiency factor. I assume it is Equ. 9 in Belyaev and Levin?

[Yes, this is correct. Equations 1-4 now describe in detail how the inlet aspiration efficiency was calculated.]

page 6 line 12: replace by "...and the LAS instrument were calculated after Brockmann (2001) on a second-by-second..."

[Done.]

page 7, line 30ff / Fig 6: What was the averaging time for the size distributions? It's hard to see from the graphs a1, b1, c1. These are not the reported 1-second data products but rather 30 second averages.

[We have redrawn this figure (see below). The averaging time has been changed 60 s and is now explicitly described in the figure caption. We have modified Fig. 6 to use the final submitted data (which eliminated some portions of data due to cloud con-tamination) and to zoom in on 10-minute segments of the flight. The size distribution plots (middle and right columns) have been modified to show the inverted and interpo-lated size distribution as a shaded histogram, which makes things clearer (especially for color-impaired readers).]

page 8, line 20: please refer to section 4.3.1 when mentioning that the time resolution is variable.

[Done.]

Section 4.3.1: looking at Fig 6, b2 and c2, the first channels of NMASS (< 15 nm) may also suffer from poor counting statistics. Why is the averaging not applied to the first

NMASS channels as well?

[Each channel of the NMASS measures the cumulative number of particles larger than a threshold diameter, so the statistics for each channel are usually excellent. The red solid lines in Fig. 6b2 and 6c2 show the differences between the channels, which may not be statistically significant when there are few particles in this size range. However, it would be challenging to combine channel-specific averaging time intervals with the algorithm that inverts all the NMASS data together (Eq. 5), and there would be physically improbable results (such as sharp edges to the size distribution) at the boundaries between averaged and non-averaged channels.]

page 9 line 18: Any literature references for typical Aitken mode size range definitions?

[We have added a reference to Whitby, 1978, who provides a thorough description of aerosol modal structure.]

page 9 line 32: The AMS measures only the accumulation mode. The density of the nucleation mode may be different (higher, if assuming that H2SO4 plays an important role for nucleation).

[It is correct that we don't know the composition of the nucleation mode. The AMS does capture the composition of particles in the 25-60 nm range of the Aitken mode. However, their composition is averaged with that of the accumulation mode, which often dominates. Thus, estimating the density requires making assumptions about the composition of the smaller particles, but both poorly neutralized sulfate and organics likely play a role. Applying the average composition of the non-refractory component of the Aitken+accumulation modes to these smaller particles is the most impartial assumption we can make.]

page 11 line 32: remove line break.

[Corrected.]

page 12 line 14: this is channel NMASS-1 CPC5 in Figure 2, right?

[Correct, and now we note this in the text.]

page 12, line 35-36: That would mean that the inlet transmission is unity up to 3 _m and drops immediately to zero. On page 5 it was said that at 12 km the transmission at 3 _m is still 50%. Would it be possible to average the >7 km distribution over a longer time? What does "> 7 km" mean exactly? Can you give the altitude range that is averaged here?

[We expect a more gradual decline in the inlet transmission than an abrupt drop, but don't have the measurements to support it. Unfortunately, there is no underlying theoretical or experimental basis for particle losses in a turbulent, conical diffuser on which to base a more sophisticated expectation. The CAS probe on the CAPS has poor sizing resolution in this Mie resonance range (and the LAS is not great, either). The LAS is also very limited by its low sample flow rate (<1 cmˆ3 per s), which leads to poor statistics. Altogether, this makes it difficult to provide a more quantitative basis for evaluating inlet performance, other than to say it appears to have reasonable transmission up to 3 $\mu$m at high altitude, and to the maximum size considered, 4.8 $\mu$m, at low altitude. We see that sampling efficiency falls (apparently rapidly) at larger sizes. We prefer to leave the discussion as it stands now–generally consistent with the findings of McNaughton et al. (2007). The altitude range of the averaging for panel b (7 to 12.9 km; now noted in the figure caption) included all available data for this flight in that altitude range and at RH<35%, so there is no further help for the statistics.]

[Figure caption for modified Fig. 6 (below):

Figure 6. (a1, b1, c1) Left column shows three example time plots of STP concentrations measured by each channel of the two NMASS instruments and by the non-thermodenuded UHSAS on 13 February 2017 on a flight over the South Atlantic Ocean between Punta Arenas, Chile (43.2° S, 70.9° W), and Ascension Island, UK (7.9° S, 14.3° W). Each plot shows 10 minutes of flight time and uses the same scales, and the cases were chosen to show how size distributions dominated by different modes

are measured. "Striping" of the UHSAS concentrations in (b1) and (c1) is caused by poor 1-s counting statistics at low concentrations. (a2, b2, c2) Center column shows linear particle number size distributions averaged over the 1-minute interval shown by the grey shading in (a1), (b1), and (c1), respectively. These size distributions show raw NMASS size distributions (dashed red curve) obtained by simple differencing of adjacent CPC channels using the 50% efficiency values of each CPC response curve (Fig. 2). Negative differences between the channels are shown as gaps in the curve. The solid blue curve shows the raw UHSAS size distribution, which is reported in 99 channels. The shaded histogram shows the combined final data, composed of the inverted NMASS data and the time-averaged and size-interpolated UHSAS and LAS data as reported in the ATom archive using 20 constant logarithmic size bins per decade of diameter. (a3, b3, b3) Right column shows the same size distributions plotted using a logarithmic y-axis to illustrate the accumulation mode better.]

―――――――――――――

**Fig. 1.**

---

## Author Comment (AC2) · 8 May 2019

We thank both referees for their thorough and helpful reviews of this manuscript. We respond to each comment below in brackets.

Reviewer #2

GENERAL COMMENT The manuscript describes with large richness of detail the methods applied for obtaining particle size distributions from the aerosol instrumentation deployed during ATom 1 to 4 on the DC-8, including the uncertainties associated to the parameters and derived data products, such as the aerosol scattering coefficient

or the aerosol mass concentration. The applied methods and associated uncertainties are described very carefully, whereas in-depth intercomparisons between different instruments and, wherever appropriate, between instrument parameters and data products demonstrate the high data quality of the ATom data set. The manuscript will serve as the reference document for any scientific analysis based on the ATom aerosol data set. It is therefore of high relevance for the scientific community and will set a standard for future research campaigns using instrument combinations. The manuscript fits perfectly into the scope of AMT. It is clearly structured, well organized and very well written. Only technical corrections are required before the manuscript is acceptable for publication in AMT.

[Thank you for your positive comments regarding the manuscript.]

MINOR COMMENTS 1. Page 2, line 6: the authors refer sometimes to "the paper", sometimes to "the manuscript". I suggest using "the manuscript" throughout the text.

[Done.]

2. In the abstract, the authors specify the range of particle measurements from 2.7 nm to 4.8 diameter (page 1, line 18), whereas in the instrumentation section they specify the range as from 3 nm to 930 _m diameter (page 2, line 17). These different ranges are caused by instrument specificities, but later in the manuscript the authors never used the range from 4.8 _m and larger. It might be worthwhile to state this in the abstract.

[The text has been modified to ensure that the range of size distributions (2.7 to 4.8 $\mu$m dry diameter, and up to 930 $\mu$m including the near-ambient cloud probes) is described consistently.]

3. On page 3, line 9, the authors may add "this refractive index range 'of the atmospheric aerosol' ".

[Done.]

4. On page 4, line 26, there is a typo "located 'aft' of the UH/LARGE inlet." Please correct.

[This is correct usage of the nautical/aeronautical term "aft", meaning "at, near, or toward the stern of a ship or tail of an aircraft."]

5. On page 5, line 5, the authors introduce filter samples collected during the flight which were used for post-flight chemical analyses. Please state the sampling time and the resulting spatial resolution.

[This information has been added to the manuscript. Sampling times varied, but usually ranged from ~4 to ~20 minutes; roughly 2-10 km vertically and 50-250 km horizontally.]

6. On page 6, line 15ff, the authors mention briefly the flow through the CAPS instrument and refer to Spanu et al. (2019) for the flow-induced errors in aerosol. Compared to the detailed discussion of the other instruments' uncertainties, a few more details would be good. In particular, how does the analysis of corrections published by Spanu et al. (2019) compare to the recently published detailed study on the thermodynamic correction of particle concentrations measured by underwing probes on fast-flying aircraft (Weigel et al., 2016)?

[The analyses are different, because Spanu et al. evaluated the effects on the flow caused by both the probe housing and the aircraft wing using CFD, while Weigel et al. focused on the probe only, primarily using thermodynamic calculations. There is currently a discussion on the Spanu et al. manuscript at AMTD regarding the differences between these approaches. However, because the CAPS measurements are only a minor part of the current manuscript, which focuses on the dry size distribution measurements, we wish to avoid a digression into the differences between the two studies. We therefore will keep the original phrasing of the current manuscript unchanged.]

7. On page 11, line 32, there is an erroneous line break inserted.

[Corrected.]

8. In Fig. 1, neither the track for Atom-4 is shown nor is an explanation given why this is not the case. Please specify.

[At the time of submission the ATom-4 data had not yet been finalized and publicly released. We now include the ATom-4 flight track on Fig. 1.]

9. In Fig. 7, the y-axis title is missing for the left panel.

[Thanks for noticing this error caused by cropping; corrected.]

10. In Fig. 10, the title of the y-axis is too close to the axis labels for both panels.

[Corrected.]

---

## Referee Comment (RC3) · Anonymous Referee #1 · 9 May 2019

Dear Dr. Brock, thank you very much for the detailed response to my review. However, it seems that you missed the first three comments (on the first page of my review)?

These were:

page 4 line 6-7: how many size classes can the POPS determine? page 4 lines 8-15: in which format will PALMS data by made public on the data base? page 4, line 25: HIMIL and UH/LARGE acronyms are not explained (appear here for the first time).

Could you please reply to these comments as well? Thanks.

---

## Author Comment (AC3) · 9 May 2019

My apologies, I somehow managed to drop responses to your first three questions/comments.

page 4 line 6-7: how many size classes can the POPS determine?

[The POPS reported 13 size classes between 0.18 and 3.6 $\mu$m diameter. This will be noted in the revised manuscript.]

page 4 lines 8-15: in which format will PALMS data by made public on the data base?

[All the data from the ATom projects are reported in ASCII files using the ICARTT for-

mat. PALMS data take the form of number fractions of the different particle types (e.g., sea salt, sulfate/organic/nitrate mixtures, biomass burning particles, etc.) as well as mass fractions for some specific parameters (e.g., organic-to-sulfate mass fractions). We do not plan to mention these details in the text. Please note Froyd et al (2019), currently under review at AMTD, provide extensive details on the PALMS data processing and additional size-resolved data products.]

page 4, line 25: HIMIL and UH/LARGE acronyms are not explained (appear here for the first time).

[The acronyms UH and LARGE are defined on p. 2, in the first paragraph of Sect. 2. HIMIL stands for HIAPER Modular Inlet, and HIAPER stands for High-performance Instrumented Airborne Platform for Environmental Research. These acronyms will defined in the revised manuscript where HIMIL is first used.]